# Negative Dielectric Anisotropy Liquid Crystal with Improved Photo-Stability, Anti-Flicker, and Transmittance for 8K Display Applications

**DOI:** 10.3390/molecules27217150

**Published:** 2022-10-22

**Authors:** Haiguang Chen, Youran Liu, Maoxian Chen, Tianmeng Jiang, Zhou Yang, Huai Yang

**Affiliations:** 1School of Materials Science and Engineering, University of Science and Technology Beijing, Beijing 100083, China; 2Beijing Bayi Space Liquid Crystal Material Technology Company, Beijing 102502, China; 3Beijing Key Laboratory of Liquid Crystal Materials Analysis and Application Technology, Beijing 102502, China; 4Department of Materials Science and Engineering, College of Engineering, Peking University, Beijing 100871, China; 5Key Laboratory of Polymer Chemistry and Physics of Ministry of Education, Peking University, Beijing 100871, China

**Keywords:** negative dielectric anisotropy liquid crystals, high-resolution display, photo-stability, anti-flicker, transmittance

## Abstract

Video systems such as 8K displays can provide a strong sense of presence and reality due to their extremely high resolution and wide field of view. However, high-resolution displays generally suffer from reduced transmittance, which requires the use of liquid crystals with high transmittance and high stability. In this study, negative dielectric anisotropy liquid crystal compositions with excellent photo-stability, anti-flicker capability, and high transmittance are developed, showing potential for 8K display applications. The stability of different types of negative dielectric anisotropy liquid crystal compounds is assessed under light, and the high photo-stability compounds are obtained. In addition, it is demonstrated that the flicker can be optimized from −17.6 to −47.0 by removing the compounds with a higher ion content and a larger deformation number and adding the compounds with a smaller deformation number in the negative dielectric anisotropy liquid crystal compositions. Combining with the evaluation of the factors affecting the response time, the negative dielectric anisotropy liquid crystal mixed H is designed with improved stability and flicker; thus, the response time was decreased to 9.5 ms, and the optical transmittance was 5.5% higher than that of MAT-09-1284 (for normal) and 3.1% higher than that of BY19-J01A (for 4K).

## 1. Introduction

Since the development of 8k display technology in 2011, there is a growing demand for large-size, high-resolution, and high-quality display products [1,2,3,4]. Large screen display panels with 4K and 8K resolutions [5], wide viewing angles, and wide color gamut have been developed [6,7,8].

Compared with 2K and 4K display products, the 8K display sets higher requirements for the liquid crystal materials (Table 1). Firstly, as the resolution increases from 4K to 8K, the width and length of the sub-pixel become 1/4 of the original pixel, which lowers the aperture ratio and reduces transmittance by about 40% [9,10,11,12]. Consequently, the transmittance of 8K liquid crystal material needs to be 4–6% higher than that of 2K liquid crystal material. In addition, the liquid crystals for the 8K display require better reliability (voltage holding ratio, VHR) and anti-flicker properties due to their high driving frequency.

At present, there are some solutions to improve the transmittance of LCD panels [13,14,15,16,17,18,19,20]. For liquid crystal materials, replacing positive dielectric anisotropy liquid crystals with negative dielectric anisotropy liquid crystals can increase the transmittance [21,22,23,24,25]. Unlike positive dielectric anisotropy liquid crystals, negative dielectric anisotropy liquid crystals do not change their inclination angle when they rotate in the plane under the electric field in the fringe field switching (FFS) mode (Figure 1) [26]. In addition, the reliability and flicker issues could be improved by using negative dielectric anisotropy liquid crystals. As compared to positive dielectric anisotropy liquid crystals, the disadvantages of negative dielectric anisotropy liquid crystals for display applications are summarized as follows: (1) Negative dielectric anisotropy liquid crystal molecules generally have smaller permittivity, and therefore require larger threshold voltage and exhibit larger rotational viscosity coefficient and slower response time [27,28]. (2) Negative dielectric anisotropy liquid crystals usually contain higher ionic content and therefore have lower reliability than that of positive dielectric anisotropy liquid crystals, especially after UV irradiation, which results in residual image and flicker [29,30]. The objective of this paper is to improve the stability and mitigate the flicker issue related to negative dielectric anisotropy liquid crystals by developing novel liquid crystal molecules with high transmittance suitable for 8K display applications.

## 2. Results & Discussions

### 2.1. The Reliability Improvement of Negative Dielectric Anisotropy Compounds

The reliability of negative dielectric anisotropy liquid crystals can be improved by improving the light stability of the negative dielectric anisotropy compounds and by mitigating the flicker issue of negative dielectric anisotropy liquid crystals on display.

#### 2.1.1. The Photo-Stability Improvement of Negative Dielectric Anisotropy Compounds

In this paper, the stability of different negative dielectric anisotropy compounds under light irradiation was investigated using the following method: (1) A mixed liquid crystal A was obtained by using compounds with good stability. (2) Compounds to be tested were added to mixed liquid crystal A at 10% to prepare samples for the stability measurement. (3) The prepared sample was poured into a test cell (TT1A6116), which was irradiated with a 20,000 nit backlight for 10 days. (4) The liquid crystals in the test cell were flushed with acetone and were tested using LC-MS. The composition and content of mixed liquid crystal A are shown in Appendix A, and the structures of the negative dielectric anisotropy compounds are shown in Table 2.

The experiment result showed that the Type A negative dielectric anisotropy mixture was relatively stable to light and showed similar stability regardless of the substituents. Then, the photo-stability of the Type B negative dielectric anisotropy mixture (Appendix A) was tested. The analysis of the impurities produced by the negative dielectric anisotropy compounds indicated that the impurities were phenolic substances, as shown in Figure 1.

The experiment results (Appendix A) showed that the Type B negative dielectric anisotropy compounds were relatively stable to light and showed slightly better stability than that of the Type A negative dielectric anisotropy mixture. Then, the Type C negative dielectric anisotropy mixture (Appendix A) was tested. The experiment result (Appendix A) showed that the photo-stability of the Type C negative dielectric anisotropy mixture was similar to that of Type B. Interestingly, the content of phenolic compounds increased with the increasing number of carbon atoms of alkyl and alkoxy substituents.

Subsequently, the photo-stability of the Type D negative dielectric anisotropy mixture (Appendix A) was tested. The experiment result (Appendix A) showed that the Type D negative dielectric anisotropy mixture had poor photo-stability, which was attributed to the substituents. The compounds with a biphenyl structure showed poor photo-stability due to their conjugated structure, which facilitated the formation of phenolic compounds. In addition, the effect of the substituent was more significant for biphenyl-containing compounds. Next, we synthesized Type E compounds with different bridge bonds (Appendix A), such as methoxy and ethyl, and investigated their stability upon light irradiation.

The experiment result (Appendix A) shows that when a methoxy bridge bond is introduced, the stability of compounds to light is generally poor, while when an ethyl bridge bond is introduced, the stability of photo-stability is greatly improved, but poor compared with that without a bridge bond. Next, the negative dielectric anisotropy compounds of class f (Appendix A) with the thiophene group were tested.

The experiment result (Appendix A) showed that the photo-stability of negative dielectric anisotropy compounds with the thiophene group varied significantly due to the structural differences. When cyclopropane was connected with methoxy, the compound exhibited good stability, whereas when alkenyl was introduced, the photo-stability decreased greatly with complex decomposition components. Based on the experiment results, it was concluded that the photo-stability of Types A, B, and C negative dielectric anisotropy compounds was relatively good, while the photo-stability of Types D and E negative dielectric anisotropy compounds was poor.

We further analyzed the mechanism of the decomposition of negative dielectric anisotropy compounds into phenols under light. An anti-light stabilizer was added for comparison purposes. The experiment was designed as follows: (1) A mixed liquid crystal B was obtained by using Type A and B negative dielectric anisotropy compounds. (2) The anti-light stabilizer was added to mixed liquid crystal B at 0.5%. (3) The mixture was poured into a test cell, which was irradiated with nit backlight for 10 days. (4) The liquid crystal was flushed out using acetone and was tested using LC-MS. The experiment result is shown in Figure 2, in which the peak times of the three impurities are 19.87, 20.92, and 21.74.

The experiment result (Appendix A) showed that the addition of an anti-light stabilizer can effectively reduce the formation of phenols from negative dielectric anisotropy compounds by about 40%. We speculate that the phenols were mainly due to the formation of acidic substances by PI under harsh conditions of light and heat, which induced the negative dielectric anisotropy compounds to form phenols. Therefore, the following experiments were designed.

A: The content of phenols generated by LC under different conditions was tested, including heat, combined light, and heat, with and without PI, respectively.

B: An acid was added to the liquid crystal and the content of phenols generated by LC under different conditions was tested, including heat, combined light, and heat, with and without PI, respectively.

The experiment result (Appendix A) showed that no phenol was generated for the samples without PI regardless of conditions. Next, samples with PI were tested, and the experiment result (Appendix A) showed that in the presence of PI, phenols were produced only when light and heat existed at the same time and that there was no acid involved in the reaction process. An improved method of adding an anti-light stabilizer into the negative system was put forward, which provided the experimental basis for selecting the suitable compound type in the following development.

#### 2.1.2. The Improvement of Flicker

Flicker is mainly caused by the following two reasons: (1) Ions, in the aging process, due to the asymmetric voltage, the ions move to the PI surface and adsorb on the PI surface to form an ion electric field, resulting in residual DC voltage (RDC). This further leads to the asymmetry of positive and negative half-cycle voltage, resulting in the change of transmittance and flicker issues [31]. (2) Flexo [32,33,34,35,36], flexoelectric effect refers to the phenomenon that liquid crystal molecules self-polarize and exhibit macroscopic dipole moment due to stretching and bending deformation under the action of an external electric field. In FFS mode, there is no strict horizontal electric field, but a vertical electric field is present, which will lead to the deformation of liquid crystal molecules and produce a spontaneous polarization electric field [37,38,39]. The existence of a spontaneous polarization electric field will result in a brightness difference under the positive and negative voltage of the same value. In addition, there are differences in the arrangement of liquid crystal molecules under positive and negative frames in the process of electric field switching, which results in different brightness. The improvement of flicker can be made using the following strategies: by improving the deformation number of liquid crystal molecules by selecting a structure with less shape variable. Therefore, the obtained liquid crystal molecules will have lower deformation and a smaller flexo effect.

In this work, we sought to mitigate the flicker issue by changing the ion content (Ion) and molecular shape variable (Flexo). The flicker phenomenon is more obvious under low-frequency conditions [40,41,42,43]. Therefore, mixed liquid crystal C was used to investigate the flicker phenomenon under 1 Hz. The experiment data was tested using a comprehensive tester.

The composition and content of the mixed liquid crystal C are shown in Appendix A with the physical properties in Appendix A. The stability of the biphenyl compounds 4 and 11 was poor in the mixed liquid crystal C, which was expected to result in more ions in the mixed crystal. Therefore, biphenyl compounds 4 and 11 were removed, and the compounds shown in Figure 2 and Figure 3 were added to form mixed liquid crystal D as follows:

The experiment result (Appendix A) showed that after compounds 4 and 11 in mixed liquid crystal C were removed, the VHR (charge retention) significantly increased, and ion content significantly reduced. The transmittance and flicker of the mixed liquid crystals C and D were tested, and the results are shown in Figure 3. The flicker was calculated according to Equation (1).
Flicker (dB) = 20log{(Tmax−Tmin)/[(Tmax+Tmin)/2]}(1)

On the basis of mixed liquid crystal D, we further optimized the structure to mitigate flicker by improving flexo. The thiophene structure in the component was removed to obtain mixed liquid crystal E with the ion content shown in Appendix A. The experiment data showed that the ion content remained unchanged when the thiophene structure in mixed liquid crystal D was removed. The transmittance and flicker of mixed liquid crystal E were tested at 1 Hz and are shown in Appendix A.

The experiment result showed that under the condition of the same ion content, the difference between the two transmittance curves was significant. The flicker of mixed liquid crystal D is −26.7 and that of mixed liquid crystal E is −39.6. The removal of the thiophene structure showed an obvious positive effect on the mitigation of flicker. After removing the thiophene structure, a trifluoro substituted terphenyl structure with strong rigidity was added to the mixed liquid crystal E, as shown in Figure 4 below, to form a mixed liquid crystal system F, which has a relatively rigid benzene ring structure and is not easy to deform. The tested transmittance and flicker are shown in Appendix A.

The experimental data showed that the trifluoro-substituted terphenyl structure had a positive effect on mitigating flicker. By adding a homologue of the trifluoro-substituted terphenyl structure into the system, as shown in Figure 5, mixed liquid crystal G was obtained, showing excellent anti-flicker properties with a dB value of −47.0, as shown in Figure 4.

Then, the mixed liquid crystals G and C were tested at different frequencies. The results (Appendix A) showed that they had similar flicker at 30 Hz, but the flicker of the mixed liquid crystal G was slightly better than that of the mixed liquid crystal C at 10 Hz and was significantly better than that of the mixed liquid crystal C at 0.5 Hz.

Based on the study of the photo-stability of negative dielectric anisotropy compounds and the influence of flicker, it was concluded that the negative dielectric anisotropy compounds a, b, and c have better photo-stability, and the negative dielectric anisotropy compounds of c have better anti-flicker property.

### 2.2. The Improvement of Response Time of Negative Dielectric Anisotropy Liquid Crystal

It can be seen from Equation (2) that the decrease in response time is related to the decrease in γ and the increase in the K value [44].
*τ_d_* = 1.238×(γd^2^/K_22_π^2^)(2)
By studying the relationship between structure and properties, we expect that (Appendix A) benzene ring, dioxane ring, and alkenyl group will decrease the γ value, while cyclohexyclic, polycyclic, and especially tetracyclic will increase the K value. Therefore, the response time can be improved by designing compounds with the above characteristics into the negative dielectric anisotropy mixed crystal system.

## 3. Materials & Methods

According to the study of the properties of negative dielectric anisotropy compounds above, the mixed liquid crystal H was obtained, which has good photo-stability and anti-flicker by using the selected negative dielectric anisotropy compounds and has a lower γ1 by using the compounds with cyclohexyclic (Table 3). The mixed liquid crystal H was tested (Table 4) and was compared with the mass-produced 2K liquid crystal MAT-09-1284 and 4K liquid crystal BY19-J01A with high transmittance to verify the performance. Firstly, the photo-stability of the mixed liquid crystal H was tested [45,46], and it was found that the mixture has excellent photo-stability under the backlight of 14,000 nits for 300 h (Table 5).

The result showed that the reliability was greatly improved after the structural optimization of the photo-stability. The flicker at different times was tested under a 30 Hz drive, and the results are shown in Table 6.

The result showed that by introducing negative dielectric anisotropy compounds with better flicker into the mixed liquid crystal H, the flicker issue can be reduced significantly. Then, the response time was measured, and the results are shown in Table 7.

The test was carried out in a 3.0 μm test cell, and the experiment results showed that the response time of the mixed liquid crystal H was greatly improved after the addition of the fast response negative dielectric anisotropy compounds, which was determined to be as low as 9.5 ms. Then, the transmittance was tested as shown in Figure 5.

The results showed that the transmittance of the mixed liquid crystal H was greatly improved, which was 5.5% higher than that of MAT-09-1284 and 3.1% higher than that of BY19-J01A.

Based on the above experiments, the mixed liquid crystal H exhibited excellent properties, including 105.5% transmittance, 92.5% VHR, and −50.6 flicker shown in Table 8. The results showed that the transmittance, VHR, and flicker properties reached the requirements for 8K liquid crystal display applications.

## 4. Conclusions

In this paper, stable negative dielectric anisotropy mixtures (Type A, B, and C) were obtained and the photo-stability of these negative dielectric anisotropy mixtures was investigated. The flicker value was optimized from −17.6 to −47.0 by removing compounds with higher ion content and by adding negative dielectric anisotropy compounds (Type C). Based on the study of negative dielectric anisotropy compounds, the mixed liquid crystal H was developed with greatly improved photo-stability and flicker, which exhibited similar properties to the mass-produced MAT-09-1284 and BY19-J01A, and the response time of the mixed liquid crystal H was increased to 9.5 ms. At the same time, the transmittance of mixed liquid crystal H was 5.5% higher than that of MAT-09-1284 and 3.1% higher than that of BY19-J01A. The developed negative mixed liquid crystal H showed higher performance, making it suitable for the application in an 8K display.

## Data Availability

Data contained within the article are available from the authors.

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
