# Peer review of "Negative Dielectric Anisotropy Liquid Crystal with Improved Photo-Stability, Anti-Flicker, and Transmittance for 8K Display Applications"

_molecules, 2022, doi:10.3390/molecules27217150_

Round 1

Reviewer 1 Report

The authors have to improve upon the quality of paper. First thing is there is no negative liquid crystal, it is the dielectric anisotropy of liquid crystal is negative. They should clearly use the word negative dielectric anisotropy liquid crystal in place of negative liquid crystal. How the mixed liquid crystal H was prepared is not discussed in the paper. The quality of presentation has to be improved. There are many incomplete sentences in the manuscript. The first figure the way it has been presented is not conveying any meaningful message. In general there are many spelling mistakes in the captions of the figure. I suggest the authors to seriously look into the manuscript and present it in a readable form so as to convey their interesting results to the  scientific community working in this field.

Reviewer 2 Report

Please see my uploaded review report. 

In this manuscript, the authors investigated several negative liquid crystal compounds and mixtures with excellent photo-stability, reduced Flickering, and improved transmittance with potential applications for 8K displays. Detailed compositions and evaluation methods were described. This manuscript can be accepted for publication after following questions and comments have been addressed satisfactorily.

1.     The compounds listed in Table 2 are not entirely new. Therefore, it seems inappropriate to use novel in the title.

2.     Photostability: The authors irradiated the samples with a 20,00-nit backlight for 10 days. What is the spectral content of your employed backlight? It is known that blue light is more harmful than the green and red lights. The photostability of LCs and alignment layers (PI and SiO2) have been investigated by C. H. Wen, et al. J. Soc. Inf. Display 13, 805 (2005) and more recently by Q. Yang, et al. Crystals 10, 765 (2020). These two references will help readers to better understand this important subject.

3.     The trifluoro substituted terphenyl structure shown in Formula (5) is duplicated in (6).

4.     In Fig. 4, the added trifluoro substituted terphenyl helps reduce the flicker substantially. As discussed in Ref. 44, the image flickers are affected by the De, viscosity, and frame rate. Please explain why doping the trifluoro terphenyl can suppress the flickers. Is it due to increased viscosity or else? How does the compound concentration affect the flicker? Will adding this compound degrade the photostability because of its longer absorption band?

5.     Equation (7): A more precise formula for the response time of an FFS cell is
1.238
×(g1d2/K22p2), as described in J. Appl. Phys. 117, 203103 (2015).

6.     Tables 3-5 show the VHR, flickers, and response time of several formulated mixtures, but the key material properties are missing. Please add another Table listing the birefringence, viscosity, dielectric constants, and phase transition temperatures of these mixtures.

7.     Some typos are found, for examples, Line 41, “opening rate”; it is better known as aperture ratio. Line 44 VHR represents voltage holding ratio; Line 50: FFS. Please check the entire manuscript carefully.

Round 2

Reviewer 1 Report

The authors have addressed most of the comments of both the reviewers. The manuscript can be accepted for publication, but before that authors have to carefully correct these minor points in the manuscript. There are in some places negative mixed liquid crystal remaining in the manuscript. And there are some misspelt  words and misrepresented symbols  for eg. cyclohexyclic and in place of micrometer is given as um.  Equation 7 in current form is not correct. As suggested by the other reviewer, authors had to include a numerical factor of 1.238 to make expression  more precise,  and the rest of symbols in old version were correct. I suggest the authors to carefully go through the manuscript and correct the relevant spellings, symbols before it is published.